# Prescriptive SVD-Inspired Attention via Spectral Energy Retention

## Abstract

Self-attention is central to modern Transformer architectures, but its dense dot-product formulation makes it difficult to identify which internal directions are structurally important and which can be modified without disrupting the model. SVD-Inspired Attention (SVDA) addresses part of this problem by introducing a learned diagonal spectrum into the query-key score interaction, making latent attention directions explicitly inspectable through indicators such as spectral entropy, effective rank, sparsity, alignment, selectivity, and perturbation response. The paper examines the transition from diagnostic interpretation to operational intervention. A diagnosis–intervention–verification framework is proposed, in which the learned SVDA spectrum is used to guide targeted changes to the attention-score operator. The framework treats the learned spectrum as an intervention surface where spectral coefficients can be masked, retained, regularized, or compared across heads according to their role in score formation. This view is instantiated through spectral energy retention, which converts the learned spectrum into a score-structural intervention rule. Experiments on FashionMNIST, CIFAR-10, CIFAR-100, and Food-101 show that low-energy score directions can be removed while preserving accuracy within experimental noise and reducing the score-forming part of the attention operator. The contribution is a controlled demonstration that intrinsic spectral diagnostics can be converted into verified, structurally realizable modifications of attention, rather than remaining post-hoc descriptive indicators.

## 1 Introduction

Transformer architectures have reshaped modern machine learning by replacing fixed local processing patterns with adaptive self-attention. In vision, language, dense prediction, and multimodal learning, attention has become a dominant mechanism for routing information across tokens, spatial regions, or semantic units. Yet the same flexibility that makes attention powerful also makes it difficult to interpret. Standard scaled dot-product attention forms dense interactions between projected queries and keys, but these interactions do not explicitly separate direction, magnitude, spectral concentration, or latent dimensional usage. As a result, attention maps may be visually inspectable but structurally opaque.

SVD-Inspired Attention (SVDA) was introduced as a geometrically structured alternative to standard dot-product attention (Arampatzakis et al., 2025). Its core idea is to replace unconstrained query-key interaction with a formulation in which row-normalized query/key projections interact through a learned diagonal spectral modulation matrix. This structure draws inspiration from the singular-value decomposition, where directional information and spectral importance are represented separately. In subsequent work, SVDA was applied to monocular depth estimation (Arampatzakis et al., 2026b) and image classification (Arampatzakis et al., 2026a), showing that the same spectral formulation can support intrinsic attention diagnostics across both dense prediction and classification settings. The central value of SVDA in these studies was *diagnostic*. The learned spectrum made it possible to monitor spectral entropy, effective rank, spectral sparsity, angular alignment, attention selectivity, and perturbation robustness across layers, heads, and training epochs. These quantities provided a structured description of how attention organizes internally. They exposed whether a head used many or few latent directions, whether spectral energy was concentrated or diffuse, whether attention became more selective with depth, and whether attention distributions remained stable under

small perturbations. However, diagnostic interpretability leaves an important question unanswered. Weak spectral directions, redundant heads, low-effective-rank layers, and perturbation-unstable attention maps should not remain merely descriptive diagnostics. They should provide evidence for targeted compression, pruning, regularization, or architectural adjustment. *Interpretability becomes more useful when it supports intervention.* A diagnostic should not only describe model behavior, but ideally, it should also indicate where compression, pruning, regularization, or architectural adjustment may be justified.

In this work, *prescription* means evidence-guided intervention rather than automatic optimization. SVDA identifies where intervention is plausible, while empirical verification remains necessary. A prescriptive extension of SVDA is developed, in which spectral diagnostics are used to guide targeted interventions on the attention-score operator. The term prescriptive is used carefully. We do not claim that a particular spectral pattern is universally optimal, nor that lower entropy, lower rank, or higher sparsity is always desirable. Instead, the explicit spectral structure of SVDA is treated as evidence for controlled intervention. Since the diagonal entries of the learned modulation matrix directly scale latent attention directions, modifying them induces direct and analyzable changes in the attention scores. The proposed framework follows a *diagnosis–intervention–verification* loop. First, an SVDA model is trained and its spectral structure is diagnosed. Second, interventions are proposed based on the observed spectral behavior, where weak directions may be pruned, consistently low-rank heads may be compressed, head redundancy may be identified as a candidate target for further intervention analysis, and unstable heads or layers may be targeted for robustness-oriented fine-tuning. Third, the intervened model is verified in terms of predictive performance, compression, robustness, efficiency, and attention-structure indicators. The *theoretical contribution* of this paper is to show that such interventions can be defined formally as transformations of the SVDA spectrum. In particular, we show (a) that threshold pruning of small spectral entries produces bounded perturbations in pre-softmax attention scores, (b) that effective rank provides a continuous, bounded measure of active spectral dimensionality, (c) that thresholded spectra define explicit compressed support and (d) that perturbation sensitivity is controlled in part by the spectral norm of the learned modulation matrix. These results establish a mathematical basis for SVDA-guided intervention before empirical validation. The *empirical validation* in this paper focuses on SVDA-based image classification, where the learned spectrum is used to construct score-compressed attention modules. Experiments on FashionMNIST, CIFAR-10, CIFAR-100, and Food-101 test whether an energy-retention prescription can remove low-energy score directions while preserving accuracy, reducing parameters, and reducing estimated MACs. Food-101 is included as a larger natural-image validation setting with 224×224 inputs and 196 visual tokens. Matched random and largest-$\Sigma$ controls are included to distinguish spectrum-guided intervention from arbitrary or destructive pruning.

The original SVDA formulation introduced geometry into attention. The present work examines whether that geometry can guide controlled modification of the attention-score pathway. The main contributions are listed below.

- A diagnosis–intervention–verification framework is introduced for SVDA, moving from passive inspection of attention spectra to actionable model intervention.

- Spectral-direction pruning, effective-rank control, sparsity-based compression, robustness-guided intervention, and head-redundancy analysis are formalized as transformations of the SVDA attention operator.

- Elementary but useful bounds are provided showing that pruning small spectral directions induces bounded perturbations of pre-softmax attention scores, and that SVDA effective rank and spectral sparsity provide mathematically grounded indicators of compressibility.

- An energy-retention prescription is empirically validated, converting the learned SVDA spectrum into score-structural compression and showing preserved accuracy with reduced parameters and estimated MACs across four image-classification datasets, including a 224×224 Food-101 setting.

- SVDA is positioned as a framework in which interpretability becomes operational, and attention is not only inspected after training, but reshaped according to its learned spectral organization.

## 2 Related Work

### 2.1 Attention Interpretability and Intrinsic Explanations

The interpretability of attention mechanisms remains a central issue in Transformer research. Although attention maps are often visualized as explanations, several works have cautioned that raw attention weights should not automatically be treated as faithful explanations of model behavior (Wiegreffe & Pinter, 2019; Bastings & Filippova, 2020). In vision Transformers, Chefer et al. (2021) proposed relevance propagation beyond direct attention visualization, while Raghu et al. (2021) analyzed how representations and attention evolve across depth. More recent interpretability-oriented approaches, such as prototype-based Vision Transformers (Ma et al., 2024) and interpretability-aware ViTs (Qiang et al., 2023), attempt to make visual recognition more transparent through architectural or training-objective modifications. These approaches are important, but they either explain an already-trained attention mechanism or add auxiliary interpretability structures around it. SVDA follows a different route. The original SVDA formulation modified the attention operator itself by introducing row-normalized query/key projections and a learned diagonal spectral modulation matrix (Arampatzakis et al., 2025). Subsequent SVDA studies demonstrated that this intrinsic structure exposes diagnostic indicators in both monocular depth estimation (Arampatzakis et al., 2026b) and image classification (Arampatzakis et al., 2026a). The present work extends this line by asking whether those diagnostics can guide concrete interventions on the model.

### 2.2 Structured, Low-Rank, and Spectral Attention

Several Transformer variants introduce structure into attention for efficiency, stability, or improved representation. Linformer (Wang et al., 2020), Performer (Choromanski et al., 2021), Nyströmformer (Xiong et al., 2021), and Linear Transformers (Katharopoulos et al., 2020) reduce the cost of attention through low-rank, kernel, or linearized approximations. Sparse and routing-based methods, including Sparse Transformers (Child et al., 2019), Reformer (Kitaev et al., 2020), BigBird (Zaheer et al., 2020), Routing Transformers (Roy et al., 2021), and adaptively sparse Transformers (Correia et al., 2019), restrict token interactions to improve scalability. Closer to SVDA are approaches that introduce spectral or geometric structure into attention. Singularformer (Wu et al., 2023) decomposes attention to reduce complexity, Primal-Attention (Chen et al., 2023) uses asymmetric kernel SVD in primal representation, CosFormer (Qin et al., 2022) replaces softmax attention with cosine-based structure, and SpecFormer (Bo et al., 2023) incorporates spectral decomposition ideas in Transformer-like architectures. NormFormer (Shleifer et al., 2021) adds normalization to improve Transformer training stability. These methods demonstrate the value of structured attention, but they primarily target efficiency, stability, or representational improvement.

Several recent works have also explored more explicit forms of spectral control in Transformer architectures. Dynamic Spectral Weighting modulates attention heads using spectral characteristics of their outputs (Huang, 2025). Spectral Conditioning of Attention studies the Jacobian conditioning of attention blocks and modifies spectral properties of attention layers to improve stability and performance (Saratchandran & Lucey, 2026). Frequency-domain spectral attention methods filter the attention score matrix through learnable spectral masks (Huang, 2026). These works indicate a broader interest in using spectral structure not only for analysis, but also for controlling attention behavior. However, they differ from SVDA in that SVDA introduces an explicit learned diagonal spectrum inside the query-key interaction itself, making latent directional importance directly inspectable and intervention-ready.

### 2.3 SVDA as Diagnostic

SVDA defines the attention operator as

$$A = \text{softmax}\left(\frac{(Q\Sigma)\,K^\top}{\sqrt{d_k}}\right) \tag{1}$$

where $Q, K \in \mathbb{R}^{n \times d_k}$ are row-normalized query and key projections, and $\Sigma = \text{diag}(\sigma_1, \dots, \sigma_{d_k})$ is a learnable diagonal modulation matrix. This formulation separates directional alignment from spectral importance and

enables indicators such as spectral entropy, effective rank, spectral sparsity, angular alignment, selectivity, and perturbation robustness. The role of $\Sigma$ is to reweight latent attention directions before the query-key interaction is converted into an attention distribution. The monocular depth estimation study integrated SVDA into DPT and showed that the same indicators expose depth-wise and training-time organization of attention in dense prediction (Arampatzakis et al., 2026b). The image-classification study adapted SVDA to ViTs and used the indicators to analyze attention structure across FashionMNIST, CIFAR-10, CIFAR-100, and ImageNet-100 (Arampatzakis et al., 2026a). These studies established SVDA as an intrinsically diagnostic attention mechanism. They showed that the learned spectrum is not merely an additional parameterization, but a compact representation of how attention allocates latent directional capacity. The present work builds directly on this diagnostic foundation. Its novelty is not another application of SVDA, but the transition from diagnosis to intervention. Once weak directions, redundant heads, low-rank behavior, or unstable attention patterns are identified, the model can be modified in a structured way.

### 2.4 Transformer Pruning, Compression, and Head Redundancy

Model pruning and compression aim to reduce parameter count, computation, or memory cost while preserving predictive performance. In Transformers, attention heads, tokens, projection dimensions, and intermediate representations are natural targets for pruning. Early work showed that many attention heads can be removed with limited loss in performance (Michel et al., 2019; Voita et al., 2019). In vision Transformers, token pruning and adaptive computation methods such as DynamicViT (Rao et al., 2021), EViT (Liang et al., 2022), and TokenLearner (Ryoo et al., 2021) reduce redundant token processing. More recent pruning frameworks use stronger sensitivity or curvature information. HEART-ViT, for example, uses Hessian-guided dynamic attention and token pruning to jointly reason about token and head redundancy (Uddin et al., 2025). More broadly, adversarial pruning studies show that compression and robustness can be treated jointly, with pruning decisions evaluated not only by accuracy retention but also by their effect on robustness under adversarial perturbations (Piras et al., 2025). The present work is related to this pruning literature but differs in its source of evidence. Standard pruning methods often use magnitude, sensitivity, loss approximation, or learned gates. SVDA-guided pruning uses the learned attention spectrum itself. Since each diagonal entry of $\Sigma$ directly scales one latent attention direction, spectral sparsity and effective rank provide operator-level evidence about which directions are weak, redundant, or compressible. Thus, SVDA-guided pruning is not blind pruning applied after training, but intervention guided by the internal spectral organization of the attention mechanism.

### 2.5 Robustness and Stability of Attention

Robustness in attention-based models concerns the stability of predictions and internal representations under input perturbations, adversarial changes, or distribution shifts. Prior work has studied adversarially robust attention (Kitada & Iyatomi, 2021), semantic perturbation robustness (Munakata et al., 2022), and fine-grained sensitivity of dot-product self-attention (Havens et al., 2024). In the SVDA line, perturbation robustness was introduced as an attention-level diagnostic measuring the Frobenius change in attention maps under small input perturbations (Arampatzakis et al., 2025). The same diagnostic was later used in depth estimation and classification to inspect stability across layers and epochs (Arampatzakis et al., 2026b;a). This paper treats robustness diagnostics as actionable signals. If a layer or head shows high perturbation response, SVDA makes it possible to inspect whether instability is associated with the learned spectrum, projection geometry, or attention distribution. The theoretical analysis below further shows that attention-score sensitivity is controlled in part by the spectral norm of $\Sigma$, giving a formal basis for robustness-guided intervention.

## 3 From Diagnostic to Prescriptive SVDA

SVDA made it possible to reveal the internal structure of the Attention mechanism. The present work addresses the use of this revealed internal structure as evidence for controlled intervention. We argue that the explicit spectral parameterization of SVDA makes attention not only diagnosable but also actionable. Since the diagonal entries of $\Sigma$ determine the contribution of individual latent directions, interventions on

$\Sigma$ induce direct and analyzable changes in the attention operator. This observation motivates a prescriptive extension of SVDA, in which spectral diagnostics are used to guide pruning, compression, robustness tuning, and architectural adaptation.

Importantly, the term *prescriptive* is not used here to imply that a single spectral pattern is universally optimal. Low entropy, low effective rank, or high sparsity are not inherently desirable in all layers, tasks, or training scenarios. Rather, prescriptive SVDA means that the spectral diagnostics provide concrete evidence for targeted intervention. A low-energy spectral direction may be pruned, head redundancy may be flagged for further analysis, an unstable layer may be regularized or fine-tuned, and a persistently low-rank region of the model may justify architectural compression. The proposed framework therefore follows a simple diagnosis–intervention–verification loop:

$$\text{train} \longrightarrow \text{diagnose} \longrightarrow \text{intervene} \longrightarrow \text{verify}$$

The central theoretical claim developed in this section is that SVDA-guided interventions are not arbitrary engineering operations. Because $\Sigma$ enters the attention score matrix explicitly and diagonally, several common interventions can be defined as formal transformations of the SVDA spectrum and therefore of the induced attention operator. In particular, pruning small spectral directions, constraining effective rank, and controlling perturbation sensitivity can be related to bounded changes in the pre-softmax scores and, consequently, to bounded changes in the attention distribution.

### 3.1 The SVDA Spectrum as a Control Surface

Let

$$S_\Sigma = \frac{Q\Sigma K^\top}{\sqrt{d_k}} \tag{2}$$

denote the pre-softmax SVDA score matrix. The attention matrix is then

$$A_\Sigma = \text{softmax}(S_\Sigma) \tag{3}$$

where the softmax is applied row-wise.

Because $Q$ and $K$ are row-normalized, each query vector $q_i$ and key vector $k_j$ satisfies

$$\|q_i\|_2 = \|k_j\|_2 = 1 \tag{4}$$

The corresponding score between tokens $i$ and $j$ is therefore

$$S_{\Sigma,ij} = \frac{1}{\sqrt{d_k}} q_i^\top \Sigma k_j = \frac{1}{\sqrt{d_k}} \sum_{r=1}^{d_k} \sigma_r q_{ir} k_{jr} \tag{5}$$

where equation 5 makes explicit why $\Sigma$ can be treated as a control surface. Each $\sigma_r$ modulates the contribution of one latent direction to every query-key score. Increasing, shrinking, masking, or regularizing $\sigma_r$ has a direct effect on the attention operator. This is different from post-hoc interpretation of a trained attention map, as in SVDA, the interpretable structure is part of the operator itself.

We therefore define an *SVDA spectral intervention* as any transformation

$$\mathcal{T} : \Sigma \mapsto \widetilde{\Sigma} \tag{6}$$

that replaces the learned spectrum $\Sigma$ by a modified spectrum $\widetilde{\Sigma}$, producing the intervened attention operator

$$A_{\widetilde{\Sigma}} = \text{softmax}\left(\frac{Q\widetilde{\Sigma}K^\top}{\sqrt{d_k}}\right) \tag{7}$$

Examples include threshold pruning, rank-targeted modulation, sparsity-inducing shrinkage, and robustness-oriented spectral smoothing. The remainder of this section formalizes these interventions and provides elementary guarantees that connect them to the behavior of the attention operator.

### 3.2 Spectral Pruning

A direct use of the learned SVDA spectrum is to identify latent directions whose contribution to attention is negligible. Given a threshold $\tau > 0$, a binary mask is defined for each spectral direction $r = 1, \ldots, d_k$ as

$$m_r^{(\tau)} = \begin{cases} 1, & |\sigma_r| \geq \tau \\ 0, & |\sigma_r| < \tau \end{cases} \tag{8}$$

and the pruned spectrum

$$\Sigma_\tau = \operatorname{diag}\left(m_1^{(\tau)}\sigma_1, \ldots, m_{d_k}^{(\tau)}\sigma_{d_k}\right) \tag{9}$$

The resulting pruned SVDA attention is

$$A_{\Sigma_\tau} = \operatorname{softmax}\left(\frac{Q\Sigma_\tau K^\top}{\sqrt{d_k}}\right) \tag{10}$$

The intuition is simple: if a spectral direction has very small magnitude, then removing it should only weakly perturb the attention scores. The following result makes this precise at the pre-softmax level.

**Theorem 1** (Bounded score perturbation under spectral pruning). *Let $Q, K \in \mathbb{R}^{n \times d_k}$ have row-normalized rows, and let $\Sigma$ and $\Sigma_\tau$ be defined as in equation 9. Let*

$$S_\Sigma = \frac{Q\Sigma K^\top}{\sqrt{d_k}}, \qquad S_{\Sigma_\tau} = \frac{Q\Sigma_\tau K^\top}{\sqrt{d_k}} \tag{11}$$

*Then, for every pair of tokens $i, j$,*

$$|S_{\Sigma,ij} - S_{\Sigma_\tau,ij}| \leq \frac{\|\Sigma - \Sigma_\tau\|_2}{\sqrt{d_k}} \tag{12}$$

*Moreover, since $\Sigma - \Sigma_\tau$ is diagonal and contains only pruned entries,*

$$|S_{\Sigma,ij} - S_{\Sigma_\tau,ij}| \leq \frac{\tau}{\sqrt{d_k}} \tag{13}$$

*Proof.* For any token pair $(i, j)$, let $q_i, k_j \in \mathbb{R}^{d_k}$ denote the column-vector representations of the corresponding rows of $Q$ and $K$. Since the rows of $Q$ and $K$ are normalized, $\|q_i\|_2 = \|k_j\|_2 = 1$. The $(i, j)$ entry of the score matrix is

$$S_{\Sigma,ij} = \frac{1}{\sqrt{d_k}} q_i^\top \Sigma k_j \tag{14}$$

Similarly,

$$S_{\Sigma_\tau,ij} = \frac{1}{\sqrt{d_k}} q_i^\top \Sigma_\tau k_j \tag{15}$$

Therefore,

$$S_{\Sigma,ij} - S_{\Sigma_\tau,ij} = \frac{1}{\sqrt{d_k}} q_i^\top (\Sigma - \Sigma_\tau) k_j \tag{16}$$

By the definition of the spectral norm,

$$|q_i^\top (\Sigma - \Sigma_\tau) k_j| \leq \|q_i\|_2 \|\Sigma - \Sigma_\tau\|_2 \|k_j\|_2 \tag{17}$$

Using $\|q_i\|_2 = \|k_j\|_2 = 1$ gives equation 12. Since $\Sigma - \Sigma_\tau$ is diagonal and contains only the pruned entries, its spectral norm is the maximum absolute removed diagonal entry:

$$\|\Sigma - \Sigma_\tau\|_2 = \max_{r: m_r^{(\tau)}=0} |\sigma_r| \leq \tau \tag{18}$$

Substitution yields equation 13. $\qquad\square$

Theorem 1 provides the first formal justification for SVDA-guided pruning. It shows that pruning is controlled by the magnitude of the removed spectral directions. Thus, when the SVDA diagnostic identifies consistently weak spectral components, their removal has a bounded effect on every pre-softmax attention score. This does not by itself guarantee unchanged task accuracy, since the full network and downstream loss may amplify or compensate for such changes. It does, however, justify the intervention as a controlled modification of the attention operator rather than an arbitrary deletion.

### 3.3 Attention Perturbation After Pruning

The previous theorem bounds the perturbation of the pre-softmax score matrix. Since the attention matrix is obtained through a row-wise softmax, the same perturbation should also be related to controlled changes in the resulting attention distribution.

**Proposition 1** (Controlled attention perturbation under spectral pruning)**.** *Let $A_\Sigma$ and $A_{\Sigma_\tau}$ be the row-wise softmax attention matrices generated from $S_\Sigma$ and $S_{\Sigma_\tau}$. If no individual attention score changes by more than $\epsilon$ after pruning*

$$\|S_\Sigma - S_{\Sigma_\tau}\|_\infty \leq \epsilon \tag{19}$$

*where $\|\cdot\|_\infty$ denotes the entrywise maximum norm as*

$$\|M\|_\infty = \max_{i,j} |M_{ij}| \tag{20}$$

*then for each attention row $i$*

$$\|A_{\Sigma,i} - A_{\Sigma_\tau,i}\|_1 \leq C_{\text{sm}}\epsilon \tag{21}$$

*where $C_{\text{sm}}$ denotes a finite Lipschitz constant for the row-wise softmax under the chosen finite-dimensional norm. In particular, using the bound from Theorem 1, the change in each attention row is controlled by a quantity proportional to $\tau/\sqrt{d_k}$.*

*Proof.* The row-wise softmax is a smooth function with bounded Jacobian on finite-dimensional Euclidean space. Therefore, for each row, it is Lipschitz continuous under standard vector norms. Applying this Lipschitz property to the score rows $S_{\Sigma,i}$ and $S_{\Sigma_\tau,i}$ gives equation 21. The second statement follows from Theorem 1, which bounds the elementwise score perturbation induced by spectral pruning. □

This proposition connects the spectral pruning operation to the actual attention distribution. The result is intentionally modest, as it does not claim that pruning always improves the model, nor that the downstream prediction will be invariant. It claims only that the attention-level change induced by pruning small spectral directions is bounded and analyzable. This is precisely the kind of guarantee needed to motivate diagnosis-guided intervention.

### 3.4 Effective Rank as a Controllable Attention Dimension

SVDA also enables a continuous notion of how many latent directions are effectively used by an attention head. Given the diagonal spectrum $\Sigma = \text{diag}(\sigma_1, \ldots, \sigma_{d_k})$, define the normalized spectral energy distribution

$$p_r = \frac{\sigma_r^2}{\sum_{s=1}^{d_k} \sigma_s^2}, \qquad r = 1, \ldots, d_k \tag{22}$$

assuming $\Sigma \neq 0$. The spectral entropy is

$$H(\Sigma) = -\sum_{r=1}^{d_k} p_r \log p_r \tag{23}$$

and the effective rank is

$$r_{\text{eff}}(\Sigma) = e^{H(\Sigma)} \tag{24}$$

This quantity provides a differentiable proxy for the number of active spectral directions. It is not a hard rank in the linear-algebraic sense. Rather, it measures how concentrated or diffuse the spectral energy is.

**Proposition 2** (Bounds of SVDA effective rank). *For any nonzero diagonal spectrum $\Sigma \in \mathbb{R}^{d_k \times d_k}$,*

$$1 \leq r_{\text{eff}}(\Sigma) \leq d_k \tag{25}$$

*The lower bound is attained when all spectral energy is concentrated in a single direction. The upper bound is attained when the spectral energy is uniformly distributed across all $d_k$ directions.*

*Proof.* The vector $p = (p_1, \ldots, p_{d_k})$ defined in equation 22 is a probability distribution. Its Shannon entropy satisfies

$$0 \leq H(p) \leq \log d_k \tag{26}$$

The minimum entropy $H(p) = 0$ occurs when $p$ is a point mass, meaning all spectral energy is concentrated in one direction. The maximum entropy $H(p) = \log d_k$ occurs when $p_r = 1/d_k$ for all $r$. Exponentiating the inequality gives

$$1 \leq e^{H(p)} \leq d_k \tag{27}$$

$\square$

This result formalizes the interpretation of effective rank as an attention-dimensionality indicator. In diagnostic SVDA, $r_{\text{eff}}$ reveals whether a head uses many latent directions or only a few dominant ones. In prescriptive SVDA, the same quantity can be used to guide intervention. For example, a head with consistently low effective rank may be a candidate for dimensional compression, while a head whose effective rank collapses too early may indicate premature specialization.

A direct prescriptive extension is to introduce a target-rank objective

$$\mathcal{L}_{\text{rank}} = \left(r_{\text{eff}}(\Sigma) - r^\star\right)^2 \tag{28}$$

where $r^\star$ is a desired effective rank. This does not assert that one target rank is globally optimal. Instead, it defines a controlled mechanism for testing whether task performance and robustness improve when spectral capacity is encouraged to occupy a specified range.

### 3.5 Spectral Sparsity and Compressibility

While effective rank measures the soft dimensionality of spectral usage, spectral sparsity measures the proportion of latent directions that are effectively inactive. For a threshold $\epsilon > 0$, define

$$P_\epsilon(\Sigma) = \frac{1}{d_k} \left|\{r : |\sigma_r| < \epsilon\}\right| \tag{29}$$

The complementary quantity

$$C_\epsilon(\Sigma) = \left|\{r : |\sigma_r| \geq \epsilon\}\right| \tag{30}$$

is the number of active spectral directions above threshold. This number defines the effective compressed dimension of the head after thresholding.

**Proposition 3** (Spectral support after threshold pruning). *Let $\Sigma_\epsilon$ be obtained from $\Sigma$ by threshold pruning all entries satisfying $|\sigma_r| < \epsilon$. Then the SVDA score matrix*

$$S_{\Sigma_\epsilon} = \frac{Q\Sigma_\epsilon K^\top}{\sqrt{d_k}} \tag{31}$$

*depends on at most $C_\epsilon(\Sigma)$ latent directions.*

*Proof.* By construction, $\Sigma_\epsilon$ has nonzero diagonal entries only for indices $r$ satisfying $|\sigma_r| \geq \epsilon$. Therefore,

$$q_i^\top \Sigma_\epsilon k_j = \sum_{r:|\sigma_r|\geq\epsilon} \sigma_r q_{ir} k_{jr} \tag{32}$$

All pruned directions make zero contribution to the score. Hence, the score matrix depends only on the $C_\epsilon(\Sigma)$ surviving directions. $\square$

This result justifies the use of SVDA spectra as explicit compression indicators. If a head consistently uses only a small number of active spectral directions, the attention computation can be compressed accordingly. The proposition does not imply that the compressed model is necessarily optimal. It only shows that the compression is aligned with the actual spectral support learned by the attention operator.

### 3.6 Robustness-Guided Spectral Intervention

SVDA also provides a natural way to reason about attention robustness. Let $x$ be an input and $x + \delta$ a perturbed version of the same input. The diagnostic perturbation response of an attention head can be measured as

$$\Delta_A(x, \delta) = \|A_\Sigma(x) - A_\Sigma(x + \delta)\|_F \tag{33}$$

A large value of $\Delta_A$ indicates that the attention distribution changes substantially under a small input perturbation. This can be used diagnostically to identify unstable layers or heads. It can also motivate a targeted intervention, such as fine-tuning with a robustness penalty:

$$\mathcal{L}_{\text{rob}} = \|A_\Sigma(x) - A_\Sigma(x + \delta)\|_F^2 \tag{34}$$

The following proposition states why the magnitude of $\Sigma$ matters for robustness.

**Proposition 4** (Spectral norm control of score sensitivity)**.** *Let*

$$S_\Sigma(x) = \frac{Q(x)\Sigma K(x)^\top}{\sqrt{d_k}} \tag{35}$$

*be the SVDA score matrix. For a perturbed input $x + \delta$, assume that the normalized query and key matrices change to $Q' = Q + \Delta Q$ and $K' = K + \Delta K$. Then*

$$\|S_\Sigma(x + \delta) - S_\Sigma(x)\|_F \leq \frac{\|\Sigma\|_2}{\sqrt{d_k}} \left( \|\Delta Q\|_F \|K'\|_2 + \|Q\|_2 \|\Delta K\|_F \right) \tag{36}$$

*Thus, for fixed projection perturbations, attention-score sensitivity is controlled by the spectral norm of $\Sigma$.*

*Proof.* We write

$$S_\Sigma(x + \delta) - S_\Sigma(x) = \frac{1}{\sqrt{d_k}} \left( Q'\Sigma K'^\top - Q\Sigma K^\top \right) \tag{37}$$

Adding and subtracting $Q\Sigma K'^\top$ gives

$$Q'\Sigma K'^\top - Q\Sigma K^\top = (Q' - Q)\Sigma K'^\top + Q\Sigma(K' - K)^\top \tag{38}$$

Therefore,

$$\|S_\Sigma(x + \delta) - S_\Sigma(x)\|_F \leq \frac{1}{\sqrt{d_k}} \left( \|\Delta Q\Sigma K'^\top\|_F + \|Q\Sigma\Delta K^\top\|_F \right) \tag{39}$$

Using submultiplicativity of matrix norms,

$$\|\Delta Q\Sigma K'^\top\|_F \leq \|\Delta Q\|_F \|\Sigma\|_2 \|K'\|_2 \tag{40}$$

and

$$\|Q\Sigma\Delta K^\top\|_F \leq \|Q\|_2 \|\Sigma\|_2 \|\Delta K\|_F \tag{41}$$

Combining the two inequalities proves equation 36. $\qquad\square$

This proposition supports robustness-guided intervention. If a layer shows high perturbation response, the cause may lie in unstable projections, excessive spectral amplification, or both. SVDA makes this distinction inspectable because $\Sigma$ is explicit. A robustness intervention may therefore target either the spectral norm of $\Sigma$, the stability of the projections, or the affected heads/layers.

### 3.7 Head Redundancy Through Spectral Similarity

SVDA also provides a compact representation of each attention head through its learned spectrum. For layer $\ell$ and head $h$, let

$$\boldsymbol{\sigma}_{\ell,h} = (\sigma_{\ell,h,1}, \ldots, \sigma_{\ell,h,d_k}) \tag{42}$$

denote the vector of spectral coefficients. A simple spectral redundancy score between two heads $h_1$ and $h_2$ in the same layer is

$$R_\ell(h_1, h_2) = \frac{|\boldsymbol{\sigma}_{\ell,h_1}|^\top |\boldsymbol{\sigma}_{\ell,h_2}|}{\|\boldsymbol{\sigma}_{\ell,h_1}\|_2 \|\boldsymbol{\sigma}_{\ell,h_2}\|_2} \tag{43}$$

High spectral similarity does not prove functional equivalence, because heads also differ in their projections and value transformations. However, it provides a necessary diagnostic signal by which if two heads have highly similar spectra and similar attention outputs, this may indicate a candidate for further redundancy analysis.

**Proposition 5** (A sufficient condition for identical head outputs)**.** *Consider two SVDA heads $h_1$ and $h_2$ in the same layer. If they have identical query, key, and value projections, identical spectra, and identical output projections, then their head outputs are identical for every input.*

*Proof.* Under the stated assumptions, both heads produce the same $Q$, $K$, $V$, and $\Sigma$ for every input. Therefore, their score matrices, attention matrices, and value-weighted outputs are identical. Since their output projections are also identical, the final head outputs are identical. $\square$

The condition in Proposition 5 is deliberately strong and primarily serves as a formal anchor. In practice, exact equality is not expected. Instead, SVDA-guided head pruning should combine spectral similarity with empirical output similarity or contribution measures. The theoretical point is that the spectrum provides an interpretable and low-dimensional signature of head behavior, making redundancy analysis more structured than blind or random head removal.

### 3.8 Summary of the Prescriptive SVDA Principle

The proposed extension of SVDA rests on a simple principle: once attention is given an explicit spectral structure, interventions on attention can be defined at the level of that structure. The diagonal spectrum $\Sigma$ is not merely a diagnostic artifact. It is part of the attention operator, and modifying it induces direct, bounded, and analyzable changes in attention scores. The main theoretical consequences are as follows:

- Threshold pruning of small spectral directions produces bounded perturbations of pre-softmax attention scores.

- Effective rank provides a differentiable measure of active spectral dimensionality, bounded between 1 and $d_k$.

- Spectral sparsity defines an explicit compressed support for each attention head.

- Perturbation sensitivity is controlled in part by the spectral norm of $\Sigma$.

- Head spectra provide compact signatures that can support redundancy analysis.

These results do not imply that spectral intervention automatically improves every Transformer. They establish that SVDA-guided intervention is mathematically well-defined and operationally meaningful because changes to the learned spectrum induce controlled changes in the attention-score operator. This motivates the empirical validation in the following section, where the framework is instantiated through spectral energy retention. In this instantiation, the learned spectrum identifies removable score directions and enables the construction of a score-compressed attention module. The choice is not incidental, since energy retention directly connects the diagnostic spectrum to a structural modification of the query-key score pathway while preserving the value pathway.

# 4 Empirical Validation: From Diagnosis to Structural Intervention

The preceding sections defined SVDA as a spectral reparameterization of self-attention in which each head is equipped with a learned diagonal modulation vector $\Sigma$. The central claim of the present work goes beyond diagnostic interpretability. The learned spectral structure should not merely describe how a trained Transformer attends, but should also prescribe how the attention operator can be reshaped. The purpose of the empirical validation is therefore to test whether the learned SVDA spectrum can be converted into a concrete structural intervention that produces a smaller attention-score operator while preserving task performance. The experiments are designed around a deliberately conservative test of whether low-energy score directions can be removed from the score-forming pathway without degrading accuracy, and whether this removal can be realized as a physically smaller score-compressed attention module rather than only as a post hoc binary mask.

The validation focuses on the learned spectral energy of the SVDA diagonal modulation. For each attention head, the learned SVDA spectrum defines an energy distribution over latent score directions. The energy of direction $r$ in head $h$ is defined as

$$E_{h,r} = \sigma_{h,r}^2 \tag{44}$$

This definition treats the learned diagonal coefficient as a directional contribution to the score-forming geometry of the head. Directions with larger $\sigma_{h,r}^2$ are interpreted as directions to which the model assigns higher spectral weight in the construction of attention scores, while directions with smaller $\sigma_{h,r}^2$ are interpreted as lower-energy score directions. Importantly, the prescription is not based on the absolute value of $\sigma_{h,r}$ and does not use a fixed numerical cutoff such as $\sigma_{h,r} < 0.90$. Instead, it uses a relative energy-retention rule. For a prescribed retention ratio $\rho \in (0, 1]$, directions within each head are sorted by decreasing energy. The retained set of score-direction indices for head $h$ is denoted by $\mathcal{K}_h \subseteq \{1, \ldots, d_k\}$ and is chosen as the smallest set satisfying

$$\frac{\sum_{r \in \mathcal{K}_h} E_{h,r}}{\sum_{r=1}^{d_k} E_{h,r}} \geq \rho \tag{45}$$

The complement $\{1, \ldots, d_k\} \setminus \mathcal{K}_h$ is then removed from the score pathway. Thus, $\rho$ is a scale-independent spectral energy-retention parameter. It specifies the fraction of learned spectral energy to preserve in each head, not a raw threshold applied to $\sigma$. This distinction is essential because the magnitude of the learned $\sigma$ values may vary with dataset, optimization trajectory, initialization, or architecture scale, whereas the retained energy fraction remains meaningful as a relative within-head prescription.

The empirical protocol separates three related but distinct objects, including (a) the original SVDA model, (b) the masked SVDA model, and (c) the score-compressed SVDA model. The original model is trained normally with the SVDA attention operator. The masked model applies the energy-retention prescription by setting the low-energy entries of the effective diagonal spectrum to zero, while leaving the original dense query, key, value, and output projections intact. This masked model is useful because it tests the functional safety of the prescription, by which if accuracy is preserved after masking, then the removed directions were not essential to the trained model's task performance. However, masking alone does not prove structural simplification, because the dense tensors and dense matrix multiplications remain present in the implementation. Therefore, a second realization is constructed, the score-compressed SVDA model. In this model, the prescribed directions are physically removed from the query projection, the key projection, and the diagonal spectral modulation. The value projection and output projection are kept full. This score-only structural realization follows the role of $\Sigma$ in the SVDA operator. The learned coefficients $\sigma_{h,r}$ weight directions used to form the query-key score matrix, not the value representation aggregated after the softmax. The energy-retention rule is therefore a prescription over score directions. The structural model removes low-energy directions from $Q$, $K$, and $\Sigma$, while keeping the value and output projections full. Compressing the value or output pathways would define a different intervention, because it would change the represented features rather than only the score geometry selected by the SVDA spectrum.

The resulting intervention can be summarized as follows. For each trained SVDA model and for each head $h$, the set $\mathcal{K}_h$ is computed from the learned $\sigma$ values using the chosen energy-retention ratio $\rho$. The query and key projection rows corresponding to the retained directions are copied into a smaller score-compressed

attention module, and the retained entries of $\sigma$ are copied into the corresponding compressed diagonal vector. The value projection and output projection are copied without structural reduction. The score-compressed model therefore computes attention scores using only the retained spectral score directions, but applies the resulting attention weights to the full value representation. This realizes the SVDA prescription as an actual structural modification of the attention module.

The theoretical bounds in Section 3 apply to the masked score operator, while the score-compressed model is evaluated empirically as a structural realization of the same prescription. To assess whether the score-compressed realization remains close to the masked prescription, an explicit masked-versus-score-compressed equivalence diagnostic is included. Let $f_{\text{masked}}(x)$ and $f_{\text{compressed}}(x)$ denote the output logit vectors produced by the masked SVDA model and the score-compressed SVDA model for an input batch $x$, respectively. For the same test batches, the relative logit discrepancy is defined as

$$\Delta_{\text{rel}} = \frac{\|f_{\text{masked}}(x) - f_{\text{compressed}}(x)\|_2}{\|f_{\text{masked}}(x)\|_2 + \epsilon} \tag{46}$$

where $\epsilon > 0$ is a small numerical stabilizer. Prediction agreement is defined as

$$A_{\text{pred}} = \frac{1}{N} \sum_{i=1}^{N} \mathbb{I} \left[ \arg\max f_{\text{masked}}(x_i) = \arg\max f_{\text{compressed}}(x_i) \right] \tag{47}$$

where $N$ is the number of evaluated test samples and $\mathbb{I}[\cdot]$ is the indicator function. These diagnostics are included because structural compression is only meaningful if the compressed model remains close to the masked intervention that was first validated functionally. A large discrepancy between the masked and score-compressed models would indicate that the structural conversion itself changed the model in a way unrelated to the SVDA prescription. Conversely, low relative discrepancy and high prediction agreement indicate that the structural model is a verified close realization of the prescribed score-direction removal.

The experiments were conducted on FashionMNIST, CIFAR-10, CIFAR-100, and Food-101 (Bossard et al., 2014). FashionMNIST, CIFAR-10, and CIFAR-100 use a compact SVDA-ViT configuration, using $32 \times 32$ inputs, patch size 4, four Transformer blocks, four attention heads, embedding dimension 256, batch size 256, and seeds 42, 43, and 44. FashionMNIST and CIFAR-10 were trained for 80 epochs, and CIFAR-100 was trained for 100 epochs. Food-101 was added as a larger natural-image validation experiment, providing $224 \times 224$ inputs, patch size 16, 196 visual tokens, four Transformer blocks, four attention heads, embedding dimension 256, batch size 64, 40 training epochs, and the same three seeds. The same score-structural evaluation logic was used for all datasets.

The primary prescription was the conservative energy-retention rule with $\rho = 0.90$, which keeps the smallest number of score directions required to preserve at least 90% of the learned spectral energy in each head. For every trained model, four quantities were measured: classification accuracy, percentage of score directions removed, reduction in trainable parameters, and reduction in estimated multiply-accumulate operations (MACs). The parameter and MAC reductions are measured relative to the original SVDA Transformer. The MAC estimate counts the projection and attention operations implied by the implemented Transformer blocks and is used as an architecture-level arithmetic estimate rather than as a hardware latency proxy. Because the structural realization compresses only the score pathway and keeps the value/output pathway full, the expected parameter and MAC reductions are modest but structurally meaningful. This is an intentional design choice. The experiment tests whether the spectral diagnosis can be translated into a meaningful structural simplification, not whether an aggressively compressed architecture can be produced by changing unrelated parts of the model.

Matched controls were also constructed. For each SVDA energy-retention prescription, two score-compressed controls were generated with the same pruning ratio. The random-matched control removes the same number of score directions randomly within the corresponding heads. It is used as a conservative sanity control, since random removal may also preserve accuracy when the pruning level is mild. The largest-$\Sigma$-matched control removes the highest-energy directions instead of the lowest-energy ones. It is used as the direct spectral-ordering test, since consistently worse behavior under largest-$\Sigma$ removal indicates that the learned spectral energy is not merely decorative but encodes useful importance information about the score directions.

Table 1: Main score-structural SVDA results for $\rho = 0.90$.

| Dataset | Dir. pruned | Acc. change | Params ↓ | MACs ↓ | Rel. L2 | Pred. agree |
|---------|-------------|-------------|----------|--------|---------|-------------|
| FashionMNIST | 23.34% | $\approx 0.00$ pp | 3.89% | 4.20% | 0.0052 | 99.87% |
| CIFAR-10 | 23.08% | $-0.01$ pp | 3.83% | 4.14% | 0.0039 | 99.83% |
| CIFAR-100 | 28.42% | $-0.02$ pp | 4.68% | 5.10% | 0.0073 | 99.13% |
| Food-101 | 48.83% | $0.01$ pp | 7.61% | 9.47% | 0.0189 | 98.26% |

The main results for $\rho = 0.90$ are shown in Table 1. Values are means over three seeds. Across all four datasets, the SVDA prescription removes a substantial fraction of score directions while preserving accuracy within experimental noise. On FashionMNIST, 23.34% of score directions are removed, producing a 3.89% parameter reduction and a 4.20% estimated MAC reduction, with effectively no accuracy loss. The relative masked-versus-compressed discrepancy is 0.0052 and prediction agreement is 99.87%, indicating that the structural realization closely follows the masked prescription. On CIFAR-10, the same prescription removes 23.08% of score directions, reduces parameters by 3.83%, reduces estimated MACs by 4.14%, and again preserves accuracy, with a reported accuracy change of $-0.01$ percentage points. The relative discrepancy is 0.0039 and prediction agreement is 99.83%. On CIFAR-100, the task is more difficult and the learned spectrum permits slightly stronger score-direction removal. That is, 28.42% of directions are pruned, giving a 4.68% parameter reduction and a 5.10% MAC reduction. Accuracy remains preserved within experimental noise, with an accuracy change of $-0.02$ percentage points. The equivalence diagnostic remains strong, with relative discrepancy 0.0073 and prediction agreement 99.13%. The Food-101 result strengthens the validation by moving from 32×32 inputs to a 224×224 natural-image setting with 196 visual tokens. Under the same $\rho = 0.90$ prescription, SVDA removes 48.83% of score directions, reduces trainable parameters by 7.61%, and reduces estimated MACs by 9.47%. The mean accuracy change is only 0.01 percentage points, while the masked-versus-compressed relative discrepancy is 0.0189 and prediction agreement is 98.26%. The absolute Food-101 accuracy is not presented as a competitive benchmark result, since the model is a compact SVDA-ViT trained from scratch rather than a large pretrained architecture. The experiment is included to test whether the prescribed structural intervention remains close to the masked prescription and preserves accuracy in a larger-token natural-image scenario. Negative accuracy changes indicate that the score-compressed model slightly outperformed the original model in the corresponding mean over seeds. These differences are within experimental noise and are interpreted as accuracy preservation rather than improvement. For transparency, the supplementary material includes the executable implementation and the numerical results used to produce the reported tables, including the original and score-compressed accuracies.

These results support the first empirical claim of the paper, that the learned SVDA spectrum can be used prescriptively to identify removable score directions, and the resulting intervention can be realized structurally. The gain is not merely conceptual. The score-compressed models contain fewer trainable parameters and require fewer estimated MACs while preserving accuracy. At the same time, the equivalence diagnostics show that the structural realization remains very close to the masked prescription. This is important because it rules out a possible alternative interpretation, by which the compressed model is not simply a different model that happens to work. It is a close structural realization of the SVDA-prescribed score-direction removal.

A second experiment tests whether $\rho = 0.90$ is an arbitrary choice or a controllable operating point. A sensitivity analysis was conducted on CIFAR-10 using $\rho \in \{0.85, 0.90, 0.95\}$ with three seeds. The results are shown in Table 2. In this Table, values are means over three seeds, and lower $\rho$ retains less spectral energy, producing stronger structural compression, while higher $\rho$ is more conservative. The expected monotonic behavior is observed. Lowering $\rho$ from 0.95 to 0.85 retains less spectral energy, removes more score directions, and yields greater structural reduction. At $\rho = 0.85$, 31.05% of score directions are removed, giving a 5.16% parameter reduction and a 5.58% MAC reduction, while accuracy changes by only 0.01 percentage points. At $\rho = 0.90$, 23.05% of directions are removed, producing a 3.83% parameter reduction and a 4.14% MAC reduction, again with no meaningful accuracy loss. At $\rho = 0.95$, the rule becomes more conservative as only 13.35% of directions are removed, producing a 2.22% parameter reduction and a 2.40% MAC reduction,

Table 2: CIFAR-10 sensitivity to the spectral energy-retention parameter $\rho$.

| $\rho$ | Dir. pruned | Params ↓ | MACs ↓ | Acc. change | Rel. L2 | Pred. agree |
|---|---|---|---|---|---|---|
| 0.85 | 31.05% | 5.16% | 5.58% | 0.01 pp | 0.0059 | 99.74% |
| 0.90 | 23.05% | 3.83% | 4.14% | $-0.02$ pp | 0.0041 | 99.83% |
| 0.95 | 13.35% | 2.22% | 2.40% | 0.00 pp | 0.0021 | 99.96% |

with no accuracy loss. The equivalence diagnostics also become increasingly tight as $\rho$ increases. Prediction agreement rises from 99.74% at $\rho = 0.85$ to 99.96% at $\rho = 0.95$, while relative discrepancy decreases from 0.0059 to 0.0021.

The sensitivity study supports the second empirical claim that $\rho$ acts as a meaningful energy-retention control parameter. It does not behave as an arbitrary threshold. Instead, it controls the compression-conservation trade-off in the expected direction. Lower values of $\rho$ produce stronger score-direction pruning and larger structural reduction, whereas higher values of $\rho$ produce weaker pruning and tighter masked-versus-compressed equivalence. The choice $\rho = 0.90$ is therefore a balanced operating point rather than a fixed numerical cutoff. It preserves most of the learned spectral energy while still removing approximately one quarter of score directions in the compact 32×32 experiments and nearly one half of score directions in the Food-101 experiment.

The empirical claim is therefore deliberately specific. The experiments do not establish broad Transformer compression, deployment-level acceleration, or systematic superiority over random pruning at conservative compression levels. They establish that the learned SVDA spectrum can be converted into a deterministic score-pathway intervention, that low-energy score directions can be removed while preserving predictive performance, that the resulting masked prescription can be realized as a smaller query–spectrum–key score pathway, and that removing high-energy directions is consistently more disruptive. This supports the interpretation of $\Sigma$ as an operational control surface for the attention-score operator rather than only as a post-hoc diagnostic object.

The empirical claim concerns structural and arithmetic complexity reduction, not wall-clock acceleration. The score-compressed realization reduces trainable parameters and estimated multiply-accumulate operations by reducing the dimensionality of the score-forming $Q\Sigma K^\top$ pathway. These reductions are reported as architecture-level complexity estimates and are independent of hardware-specific kernel efficiency. Actual latency is not used as evidence because the current prototype uses non-fused per-head operations and variable-width score pathways, which do not exploit the optimized batched matrix multiplication kernels used by dense Transformer implementations. Thus, the present results establish a reduction in score-pathway arithmetic complexity and parameter count, while deployment-level acceleration is left to hardware-aware implementations. If $d_h$ is the original per-head score dimension and $k_h = |\mathcal{K}_h|$ is the retained score dimension after energy retention, then the query-spectrum-key score computation in head $h$ is reduced from order $O(n^2 d_h)$ to $O(n^2 k_h)$, while the value aggregation and output projection remain unchanged.

## 5 Limitations

The present validation is intentionally conservative. It demonstrates that SVDA energy retention can be converted into score-pathway structural compression with preserved accuracy, but it does not establish broad Transformer compression or deployment-level acceleration. The parameter and MAC reductions are limited by design because the structural realization compresses only the query–spectrum–key score pathway and preserves the value and output pathways. This choice isolates the structural consequence of the learned SVDA spectrum, but it also limits the attainable reduction. Other formal intervention targets discussed in the framework, including effective-rank control, robustness-guided regularization, and head-redundancy analysis, are not empirically validated in this study.

The experiments use compact SVDA-ViT configurations rather than large pretrained backbones, and Food-101 is included as a larger-token natural-image validation scenario rather than as an attempt to reach

state-of-the-art Food-101 accuracy. Larger architectures, pretrained models, language tasks, and optimized fused implementations are required to assess the practical compression and acceleration potential of the same prescriptive mechanism.

The matched random and largest-$\Sigma$ controls test whether the learned spectrum provides meaningful internal ordering. They do not replace a full comparison against specialized Transformer pruning and compression methods. The random-matched results also show that conservative score-direction removal can sometimes preserve performance even without spectral ordering. The claim is therefore not that energy retention dominates random pruning in all cases, but that it provides a deterministic spectrum-grounded rule while the largest-$\Sigma$ control verifies that high-energy directions are consistently more important. Broader comparisons would require careful matching of compression budgets, implementation details, and evaluation conditions, and are left for future work.

## 6 Conclusion

This paper presented SVDA as a move from interpretable attention to prescriptive attention. The main claim was that the learned spectral structure of an attention head should not only be inspected, but also used to guide structural intervention. By introducing a learned diagonal modulation $\Sigma$, SVDA exposes score directions whose relative energy can be measured and acted upon.

The empirical results support this claim. Using a spectral energy-retention prescription with $\rho = 0.90$, SVDA removed approximately 23–49% of attention-score directions across FashionMNIST, CIFAR-10, CIFAR-100, and Food-101. The resulting score-compressed models reduced trainable parameters by approximately 3.8–7.6% and estimated MACs by approximately 4.1–9.5%, while preserving accuracy within experimental noise. The masked and score-compressed realizations also remained highly aligned, with prediction agreement above 98% in all datasets and above 99% in the compact 32×32 datasets, confirming that the structural model remains closely aligned with the prescribed spectral simplification.

The matched controls clarify the scope of the result. SVDA consistently outperformed largest-$\Sigma$ pruning, showing that high-energy spectral directions are structurally important and should not be removed. Against random-matched pruning, the conservative energy-retention rule was approximately tied on the compact datasets and slightly positive on Food-101. The result should therefore not be overstated as systematic superiority over random pruning. Its value is instead that it provides a deterministic, spectrum-grounded, structurally realizable rule for score-pathway simplification.

The CIFAR-10 sensitivity analysis further showed that the retention parameter $\rho$ controls a meaningful compression-conservation trade-off. Lower $\rho$ values produce stronger pruning and larger parameter/MAC reductions, while higher $\rho$ values produce more conservative interventions and tighter masked-versus-compressed agreement. Thus, $\rho$ is not an arbitrary threshold, but an interpretable operating parameter.

The present work does not claim deployment-level acceleration. Although the score-compressed models reduce parameters and estimated MACs, the current prototype uses non-fused per-head operations and does not reliably improve wall-clock latency. Hardware-aware kernels and optimized implementations are left for future work.

Overall, the results show that SVDA provides more than a diagnostic view of attention. Its learned spectrum can prescribe concrete score-pathway changes, yielding structurally smaller query–spectrum–key attention operators with preserved accuracy. In this sense, SVDA does not only help explain attention. It provides a controlled mechanism for reshaping the score geometry of attention.

## A Experimental Details

This appendix provides additional implementation details for the empirical validation reported in Section 4. The purpose of the experiments is not to establish state-of-the-art classification accuracy, but to test whether the learned SVDA spectrum can be converted into a verified score-structural intervention. *All reported results*

Table 3: Datasets and input scenarios.

| Dataset | Classes | Input size | Patch size | Image tokens |
|---|---|---|---|---|
| FashionMNIST | 10 | $32 \times 32$ | 4 | 64 |
| CIFAR-10 | 10 | $32 \times 32$ | 4 | 64 |
| CIFAR-100 | 100 | $32 \times 32$ | 4 | 64 |
| Food-101 | 101 | $224 \times 224$ | 16 | 196 |

Table 4: Model and training configurations used in the experiments.

| Dataset | Blocks | Heads | Embed dim. | Patch | Batch | Epochs | Seeds |
|---|---|---|---|---|---|---|---|
| FashionMNIST | 4 | 4 | 256 | 4 | 256 | 80 | 42, 43, 44 |
| CIFAR-10 | 4 | 4 | 256 | 4 | 256 | 80 | 42, 43, 44 |
| CIFAR-100 | 4 | 4 | 256 | 4 | 256 | 100 | 42, 43, 44 |
| Food-101 | 4 | 4 | 256 | 16 | 64 | 40 | 42, 43, 44 |

*are therefore evaluated relative to the original trained SVDA model under the same architecture and training configuration.*

## A.1 Datasets and Input Settings

Four image-classification datasets were used, including FashionMNIST, CIFAR-10, CIFAR-100, and Food-101. Table 3 summarizes the corresponding class counts, input resolutions, patch sizes, and image-token counts. The token count excludes any optional classification token and refers only to image patch tokens. FashionMNIST, CIFAR-10, and CIFAR-100 use compact $32 \times 32$ inputs with patch size 4, producing 64 image tokens. Food-101 is included as a larger natural-image validation scenario with $224 \times 224$ inputs and patch size 16, producing 196 image tokens. Food-101 is used to test whether the score-structural prescription remains stable in a higher-resolution natural-image setting. It should not be interpreted as a Food-101 benchmark-optimization experiment, since the model is a compact SVDA-ViT trained from scratch rather than a large pretrained architecture.

## A.2 Model Configuration

All experiments used compact SVDA-ViT configurations with four Transformer blocks, four attention heads, and embedding dimension 256. Table 4 summarizes the model and training configuration used for each dataset. The same score-compression logic was used across datasets. The Food-101 setting differs only in input resolution, patch size, batch size, number of classes, and number of training epochs. The Food-101 experiment uses a smaller batch size because of the larger $224 \times 224$ input resolution. All datasets use the same three random seeds. Reported values are means over these three seeds.

## A.3 Energy-Retention Prescription

For each attention head $h$, the learned SVDA spectrum defines a directional energy

$$E_{h,r} = \sigma_{h,r}^2.$$

For a retention ratio $\rho$, directions are sorted by decreasing energy and the smallest retained set $\mathcal{K}_h$ is selected such that

$$\frac{\sum_{r \in \mathcal{K}_h} E_{h,r}}{\sum_r E_{h,r}} \geq \rho.$$

All directions outside $\mathcal{K}_h$ are removed from the score pathway. The main experiments use $\rho = 0.90$. The sensitivity experiment on CIFAR-10 additionally evaluates $\rho \in \{0.85, 0.90, 0.95\}$. The parameter $\rho$ should not be interpreted as a raw threshold on $\sigma$. It is a relative within-head energy-retention ratio. Thus, the

same value of $\rho$ can be applied across datasets and heads even when the absolute scale of the learned spectra differs.

## A.4 Masked and Score-Compressed Realizations

The empirical protocol distinguishes three models:

- **Original SVDA model**: the trained model before intervention.

- **Masked SVDA model**: a functional intervention in which low-energy entries of the effective spectrum are set to zero while the original dense tensors remain present.

- **Score-compressed SVDA model**: a structural intervention in which the retained score directions are physically copied into smaller query and key score projections, together with the retained entries of $\Sigma$.

The score-compressed realization modifies only the score-forming pathway, namely $Q$, $K$, and $\Sigma$. The value and output projections are preserved. This follows directly from the SVDA operator, since the learned spectrum modulates query-key score formation, not the value representation being aggregated. Compressing the value or output pathways would define a different intervention and would no longer be a direct structural realization of the masked spectral-score prescription.

## A.5 Equivalence Diagnostics

To assess whether the score-compressed model remains close to the masked prescription, the logits of the masked model and the score-compressed model are compared on the same test batches. The relative logit discrepancy is

$$\Delta_{\mathrm{rel}} = \frac{\|f_{\mathrm{masked}}(x) - f_{\mathrm{compressed}}(x)\|_2}{\|f_{\mathrm{masked}}(x)\|_2 + \epsilon}$$

where $\epsilon$ is a small numerical stabilizer. Prediction agreement is

$$A_{\mathrm{pred}} = \frac{1}{N} \sum_{i=1}^{N} \mathbb{I}\left[\arg\max f_{\mathrm{masked}}(x_i) = \arg\max f_{\mathrm{compressed}}(x_i)\right]$$

Lower relative discrepancy and higher prediction agreement indicate closer agreement between the structural model and the masked intervention. These diagnostics are used to verify that the score-compressed realization remains close to the masked spectral-score prescription.

## A.6 MAC and Parameter Accounting

Parameter reductions are computed from trainable parameter counts before and after score-structural compression. MAC reductions are estimated from the projection and attention operations implied by the implemented Transformer blocks. The MAC estimate is used as an architecture-level arithmetic proxy and not as a hardware latency proxy. For a dense SVDA attention head, the score-forming projections and attention-score computation include contributions from the query projection, the key projection, the diagonal spectrum, and the query–spectrum–key matrix product. In the score-compressed realization, each head $h$ retains only $k_h = |\mathcal{K}_h|$ score directions. Thus, the score-forming part of the computation is reduced from dimension $d_h$ to $k_h$ for the $Q\Sigma K^\top$ pathway. The value and output pathways remain full by design. The total reported MAC reduction therefore reflects only the arithmetic savings induced by the smaller score pathway. It does not assume fused kernels, sparse-kernel acceleration, or deployment-level optimization. This is why wall-clock latency is treated separately as an implementation diagnostic and is not used to support the main empirical claim.

Table 5: Aggregate score-structural SVDA results.

| Dataset | Dir. pruned | Acc. change | Params ↓ | MACs ↓ | Rel. L2 | Pred. agree |
|---|---|---|---|---|---|---|
| FashionMNIST | 23.34% | $\approx 0.00$ pp | 3.89% | 4.20% | 0.0052 | 99.87% |
| CIFAR-10 | 23.08% | $-0.01$ pp | 3.83% | 4.14% | 0.0039 | 99.83% |
| CIFAR-100 | 28.42% | $-0.02$ pp | 4.68% | 5.10% | 0.0073 | 99.13% |
| Food-101 | 48.83% | $0.01$ pp | 7.61% | 9.47% | 0.0189 | 98.26% |

Table 6: Matched-control results for $\rho = 0.90$.

| Dataset | SVDA − Random | SVDA − Largest-Σ |
|---|---|---|
| FashionMNIST | near 0 pp | +1.37 pp |
| CIFAR-10 | near 0 pp | +2.08 pp |
| CIFAR-100 | near 0 pp | +4.10 pp |
| Food-101 | +0.19 pp | +4.08 pp |

Table 7: CIFAR-10 sensitivity to $\rho$.

| $\rho$ | Dir. pruned | Params ↓ | MACs ↓ | Acc. change | Rel. L2 | Pred. agree |
|---|---|---|---|---|---|---|
| 0.85 | 31.05% | 5.16% | 5.58% | $0.01$ pp | 0.0059 | 99.74% |
| 0.90 | 23.05% | 3.83% | 4.14% | $-0.02$ pp | 0.0041 | 99.83% |
| 0.95 | 13.35% | 2.22% | 2.40% | $0.00$ pp | 0.0021 | 99.96% |

## A.7   Aggregate Empirical Results

Table 5 repeats the main $\rho = 0.90$ score-structural results for completeness. Values are means over three seeds. The Food-101 original and score-compressed accuracies are approximately 41.15% and 41.14%, respectively. These absolute accuracies are reported only to document preservation under the prescribed structural intervention, not to claim benchmark competitiveness.

## A.8   Control Experiments

Two matched controls were used at the same pruning ratio as the SVDA energy-retention prescription. The random-matched control removes the same number of score directions as the SVDA prescription, but selects directions randomly within each head. The largest-Σ control removes the highest-energy score directions instead of the lowest-energy directions. Table 6 reports the matched-control results used in the main-text discussion for the $\rho = 0.90$ experiments. Values denote mean accuracy differences in percentage points. Positive values indicate that SVDA energy-retention compression gives higher accuracy than the corresponding control.

## A.9   Energy-Retention Sensitivity

The CIFAR-10 sensitivity experiment evaluates the retention parameter $\rho$ at $\rho \in \{0.85, 0.90, 0.95\}$. Table 7 reports the corresponding score-structural results.

Table 8 reports the matched-control results for the same CIFAR-10 sensitivity experiment. Values denote mean accuracy differences in percentage points.

## References

Vasileios Arampatzakis, George Pavlidis, Nikolaos Mitianoudis, and Nikos Papamarkos. Geometry meets attention: Interpretable transformers via svd inspiration. *IEEE Access*, 13:119077–119089, 2025. doi: 10.1109/ACCESS.2025.3586739.

Table 8: CIFAR-10 matched-control results.

| $\rho$ | SVDA $-$ Random | SVDA $-$ Largest-$\Sigma$ |
|---|---|---|
| 0.85 | $+0.02$ pp | $+2.32$ pp |
| 0.90 | $+0.04$ pp | $+2.18$ pp |
| 0.95 | $+0.06$ pp | $+1.15$ pp |

Vasileios Arampatzakis, George Pavlidis, Nikolaos Mitianoudis, and Nikos Papamarkos. Interpretable vision transformers in image classification via svda. *IEEE Access*, 2026a. doi: 10.1109/ACCESS.2026.3692081.

Vasileios Arampatzakis, George Pavlidis, Nikolaos Mitianoudis, and Nikos Papamarkos. Interpretable vision transformers in monocular depth estimation via svda. *Mathematics*, 14(8):1272, 2026b. doi: 10.3390/math14081272.

Jasmijn Bastings and Katja Filippova. The elephant in the interpretability room: Why use attention as explanation when we have saliency methods? In *Proceedings of the Third BlackboxNLP Workshop on Analyzing and Interpreting Neural Networks for NLP*, pp. 149–155, 2020.

Deyu Bo, Chuan Shi, Lele Wang, and Renjie Liao. Specformer: Spectral graph neural networks meet transformers. In *International Conference on Learning Representations*, 2023.

Lukas Bossard, Matthieu Guillaumin, and Luc Van Gool. Food-101 – mining discriminative components with random forests. In *Computer Vision – ECCV 2014*, volume 8694 of *Lecture Notes in Computer Science*, pp. 446–461. Springer, 2014. doi: 10.1007/978-3-319-10599-4_29.

Hila Chefer, Shir Gur, and Lior Wolf. Transformer interpretability beyond attention visualization. In *Proceedings of the IEEE/CVF Conference on Computer Vision and Pattern Recognition*, pp. 782–791, 2021.

Yingyi Chen, Qinghua Tao, Francesco Tonin, and Johan A. K. Suykens. Primal-attention: Self-attention through asymmetric kernel svd in primal representation. In *Advances in Neural Information Processing Systems*, volume 36, pp. 65088–65101, 2023.

Rewon Child, Scott Gray, Alec Radford, and Ilya Sutskever. Generating long sequences with sparse transformers, 2019.

Krzysztof Choromanski, Valerii Likhosherstov, David Dohan, Xingyou Song, Andreea Gane, Tamas Sarlos, Peter Hawkins, Jared Davis, Afroz Mohiuddin, Lukasz Kaiser, David Belanger, Lucy J. Colwell, and Adrian Weller. Rethinking attention with performers. In *International Conference on Learning Representations*, 2021.

Gonçalo M. Correia, Vlad Niculae, and André F. T. Martins. Adaptively sparse transformers. In *Proceedings of EMNLP-IJCNLP*, pp. 2174–2184, 2019.

Aaron J. Havens, Alexandre Araujo, Huan Zhang, and Bin Hu. Fine-grained local sensitivity analysis of standard dot-product self-attention. In *Proceedings of the 41st International Conference on Machine Learning*, 2024.

Z. Huang. Dynamic spectral weighting in causalselfattention. *Neurocomputing*, 2025. Early online.

Z. Huang. Spectral attention for transformers: Frequency-domain filtering of attention scores. *Artificial Intelligence Review*, 2026. Early online.

Angelos Katharopoulos, Apoorv Vyas, Nikolaos Pappas, and François Fleuret. Transformers are rnns: Fast autoregressive transformers with linear attention. In *Proceedings of the 37th International Conference on Machine Learning*, pp. 5156–5165, 2020.

Shunsuke Kitada and Hitoshi Iyatomi. Attention meets perturbations: Robust and interpretable attention with adversarial training. *IEEE Access*, 9:92974–92985, 2021. doi: 10.1109/ACCESS.2021.3092770.

Nikita Kitaev, Lukasz Kaiser, and Anselm Levskaya. Reformer: The efficient transformer. In *International Conference on Learning Representations*, 2020.

Youwei Liang, Chongjian Ge, Zhan Tong, Yibing Song, Jue Wang, and Ping Xie. Evit: Expediting vision transformers via token reorganizations. In *International Conference on Learning Representations*, 2022.

Chiyu Ma, Jon Donnelly, Wenjun Liu, Soroush Vosoughi, Cynthia Rudin, and Chaofan Chen. Interpretable image classification with adaptive prototype-based vision transformers. In *Advances in Neural Information Processing Systems*, 2024.

Paul Michel, Omer Levy, and Graham Neubig. Are sixteen heads really better than one? In *Advances in Neural Information Processing Systems*, volume 32, 2019.

Shota Munakata, Caterina Urban, Hiroshi Yokoyama, Kazuki Yamamoto, and Kazuya Munakata. Verifying attention robustness of deep neural networks against semantic perturbations. In *Proceedings of the 29th Asia-Pacific Software Engineering Conference*, pp. 560–561, 2022.

Giorgio Piras, Maura Pintor, Ambra Demontis, Battista Biggio, Giorgio Giacinto, and Fabio Roli. Adversarial pruning: A survey and benchmark of pruning methods for adversarial robustness. *Pattern Recognition*, 168:111788, 2025. doi: 10.1016/j.patcog.2025.111788.

Yao Qiang, Chengyin Li, Prashant Khanduri, and Dongxiao Zhu. Interpretability-aware vision transformer, 2023.

Zhen Qin, Weixuan Sun, Hui Deng, Dongxu Li, Yunshen Wei, Baohong Lv, Junjie Yan, Lingpeng Kong, and Yiran Zhong. Cosformer: Rethinking softmax in attention. In *International Conference on Learning Representations*, 2022.

Maithra Raghu, Thomas Unterthiner, Simon Kornblith, Chiyuan Zhang, and Alexey Dosovitskiy. Do vision transformers see like convolutional neural networks? In *Advances in Neural Information Processing Systems*, volume 34, pp. 12116–12128, 2021.

Yongming Rao, Wenliang Zhao, Benlin Liu, Jiwen Lu, Jie Zhou, and Cho-Jui Hsieh. Dynamicvit: Efficient vision transformers with dynamic token sparsification. In *Advances in Neural Information Processing Systems*, volume 34, pp. 13937–13949, 2021.

Aurko Roy, Mohammad Saffar, Ashish Vaswani, and David Grangier. Efficient content-based sparse attention with routing transformers. *Transactions of the Association for Computational Linguistics*, 9:53–68, 2021.

Michael S. Ryoo, A. J. Piergiovanni, Anurag Arnab, Mostafa Dehghani, and Anelia Angelova. Tokenlearner: What can 8 learned tokens do for images and videos? In *Advances in Neural Information Processing Systems*, volume 34, pp. 12700–12712, 2021.

Hemanth Saratchandran and Simon Lucey. Spectral conditioning of attention improves transformer performance, 2026.

Sam Shleifer, Jason Weston, and Myle Ott. Normformer: Improved transformer pretraining with extra normalization. In *International Conference on Learning Representations*, 2021.

Mohammad Helal Uddin, Liam Seymour, and Sabur Baidya. Heart-vit: Hessian-guided efficient dynamic attention and token pruning in vision transformer, 2025.

Elena Voita, David Talbot, Fedor Moiseev, Rico Sennrich, and Ivan Titov. Analyzing multi-head self-attention: Specialized heads do the heavy lifting, the rest can be pruned. In *Proceedings of the 57th Annual Meeting of the Association for Computational Linguistics*, pp. 5797–5808, 2019.

Sinong Wang, Belinda Z. Li, Madian Khabsa, Han Fang, and Hao Ma. Linformer: Self-attention with linear complexity. In *arXiv preprint arXiv:2006.04768*, 2020.

Sarah Wiegreffe and Yuval Pinter. Attention is not not explanation. In *Proceedings of EMNLP-IJCNLP*, pp. 11–20, 2019. doi: 10.18653/v1/D19-1002.

Yifan Wu, Shichao Kan, Min Zeng, and Min Li. Singularformer: Learning to decompose self-attention to linearize the complexity of transformer. In *Proceedings of the Thirty-Second International Joint Conference on Artificial Intelligence*, pp. 4433–4441, 2023. doi: 10.24963/ijcai.2023/493.

Yunyang Xiong, Zhanpeng Zeng, Rudrasis Chakraborty, Mingxing Tan, Glenn Fung, Yin Li, and Vikas Singh. Nyströmformer: A nyström-based algorithm for approximating self-attention. In *Proceedings of the AAAI Conference on Artificial Intelligence*, volume 35, pp. 14138–14148, 2021.

Manzil Zaheer, Guru Guruganesh, Avinava Dubey, Joshua Ainslie, Chris Alberti, Santiago Ontanon, Philip Pham, Anirudh Ravula, Qifan Wang, Li Yang, and Amr Ahmed. Big bird: Transformers for longer sequences. In *Advances in Neural Information Processing Systems*, volume 33, pp. 17283–17297, 2020.

