# OpenReview forum: "Prescriptive SVD-Inspired Attention via Spectral Energy Retention"
_TMLR — Under review for TMLR_

### Review · Reviewer_JiZh · 2026-06-16

**Summary Of Contributions:**

The proposed SVDA is an attention variant that puts a learned diagonal matrix inside the query-key score computation. Earlier work used it to read off diagnostics about a trained model. This paper tries to turn it into a tool for changing the model. The authors set up a loop with three steps, train, diagnose, intervene, and verify, and they treat the learned spectrum as a knob on the score-forming part of attention.
The authors also organized five kinds of intervention as edits, namely threshold pruning, effective-rank targeting, sparsity-based compression, robustness regularization, and a head-redundancy score. They prove a few small bounds that link edits to bounded changes in the pre-softmax scores.

This paper tested one intervention. The energy-retention rule keeps, in each head, the fewest score directions whose squared coefficients add up to a fraction ρ of the total, and drops the rest. The rule is built two ways, a mask that zeroes the dropped entries, and a smaller query-spectrum-key module that physically removes them. On FashionMNIST, CIFAR-10, CIFAR-100, and Food-101, setting ρ to 0.90 drops about 23 to 49 percent of score directions with no real accuracy loss, and cuts parameters by 3.8 to 7.6 percent and estimated MACs by 4.1 to 9.5 percent. Two controls are run at the same drop rate. Removing the highest-energy directions hurts accuracy a lot. Removing random directions does about as well as the proposed rule.


Strengths: the claims are scoped honestly, the split between the masked model and the compressed model is checked with logit distance and prediction agreement, and the controls are well chosen.

Weaknesses: the main selling point is weakened by the near-tie with random pruning, the gap between the masked and compressed models is never explained, per-seed variance is not given, some of the theory is trivial or empty as written, and about half the framework is never tested.

**Audience:**

Yes

**Audience Explanation:**

People working on interpretable and efficient attention care whether structure learned inside attention can do more than describe a trained model. This paper gives a clean, well-documented case of turning such structure into a checked edit of the attention operator, and it is honest about what the evidence does and does not show, including a result that nearly ties a trivial baseline. The split between the masked model and the compressed model, and the checks used to compare them, are useful and carry over to other pruning work.

**Broader Impact Concerns:**

None. The work studies interpretability-guided compression of attention on standard public image datasets. It adds no new data collection, no generative capability, and no application with foreseeable dual-use or fairness risk beyond what any image classifier carries.

**Claims And Evidence:**

No

**Claims Explanation:**

The paper hedges a lot, so at the most basic level many sentences hold up. But several of the main claims are not yet backed by clear evidence, and the gaps are specific.

First, the phrase "preserved accuracy within experimental noise" cannot be checked, because the main tables give no per-seed standard deviation, and neither do the control and sensitivity tables. With three seeds and no spread reported, the reader has to take the sub-0.02 point accuracy changes on trust. The one absolute pair the paper does give, about 41.15 and 41.14 percent on Food-101, is not enough to back the claim across four datasets and three settings.

Second, the equivalence claim sits on a gap. Masking zeroes entries. Compression removes the same directions from the query projection, the key projection, and Σ. The two score computations should come out the same, yet the reported relative logit distance reaches 0.0189 on Food-101. The likely cause is that the query and key rows are normalized again over fewer dimensions once columns are removed, which changes the geometry. If that is what happens, then Theorem 1 does not apply cleanly to the compressed model it is meant to justify.

Third, the main message claims more than the controls show. Table 6 puts the energy-retention rule near zero against random pruning on three datasets, and at plus 0.19 points on Food-101. The largest control shows only that the biggest learned coefficients matter, which is true of almost any learned parameter and would hold just as well for plain magnitude pruning of ordinary weights.

Fourth, the theory is presented as a contribution on par with the experiments, but much of it is routine or empty. Theorem 1 is Cauchy-Schwarz with unit-norm rows. Proposition 2 restates known entropy bounds. Proposition 3 holds by construction. Proposition 5 says identical heads give identical outputs, which is true by definition and supports no real redundancy test. Proposition 1 uses a softmax Lipschitz constant it never pins down, so "proportional to τ over √dk" says nothing quantitative, even though the softmax Jacobian has a known bound.

As the paper stands, several stated claims run ahead of the evidence.

**Requested Changes:**

**Critical:**

1. Find and explain the gap between the masked model and the compressed model. State plainly whether the query and key rows are normalized again over the kept dimensions after compression, and say whether Theorem 1 covers the compressed model or only the mask. If the normalization changes, fix the equivalence claim and the theory to match.

2. Report absolute accuracy with mean and standard deviation over seeds for every dataset and every setting, namely original, masked, compressed, random control, and largest-Σ control.

3. Add at least one baseline that applies the same energy or magnitude rule to the projections of a standard attention head. That shows whether the explicit spectrum is needed for the edit or whether plain magnitude pruning does the same thing.

4. Give Proposition 1 a concrete Lipschitz constant for the row-wise softmax.

**Would strengthen the work:**

5. Report how many extra parameters and how much extra compute SVDA itself adds over plain attention, including the normalization cost, so the reader can see the net effect of the 4 to 9 percent score-pathway saving.

6. Add a short comparison against a known attention or token pruning method at a matched budget, even a rough one, so the small reductions can be placed against existing work.

---

### Review · Reviewer_XroX · 2026-07-13

**Summary Of Contributions:**

### Summary

The paper extends prior SVDA attention by using the learned diagonal spectrum to prune low-energy score directions and build a compressed attention-score pathway.

### Strengths
- **Clear presentation.** The paper is generally well-written and easy to follow.
- **Simple implementation.** The pruning procedure is straightforward and easy to implement.

### Weaknesses
- **Limited novelty.** The core SVDA mechanism was introduced in prior work; this paper mainly adds a simple spectrum-guided pruning procedure.
- **Weak theory.** The theoretical results are mostly elementary norm bounds and linear algebra identities, with limited insight into why pruning should preserve task performance.
- **Narrow contribution.** The proposed diagnosis–intervention–verification framework is broad, but only one intervention, energy-based pruning, is empirically validated.
- **Modest impact.** The compression gains are small, no inference speedup is demonstrated, and comparisons to dedicated Transformer pruning methods are limited.
- **Limited empirical studies.** Experiments only consider vision transformers and studies small datasets and relatively small models.

**Audience:**

Yes

**Audience Explanation:**

Researchers interested in interpretable attention mechanisms, structured attention are likely to find this work somewhat interesting.

**Broader Impact Concerns:**

No concerns.

**Claims And Evidence:**

Yes

**Claims Explanation:**

The main claims are supported by the presented experiments and analysis. Despite having simple setups empirical results validate the proposed spectrum-guided pruning procedure. Although the theoretical claims are shallow, they are supported by the provided proofs.

**Requested Changes:**

- Expand the empirical evaluation to larger scale architectures and additional domains (e.g., language or multimodal) to demonstrate the generality of the proposed approach.

- Empirically validate additional proposed interventions (e.g., effective-rank control, robustness-guided spectral regularization, or redundancy-based compression) beyond spectrum-guided pruning.

---

### Review · Reviewer_mfo5 · 2026-07-21

**Summary Of Contributions:**

## Summary

This work builds upon SVD-Inspired Attention (SVDA), which is a formulation where the query-key score is modulated by a learned diagonal matrix $\Sigma$. The main goal of this work is to upgrade SVDA from a diagnostic mechanism to a "prescriptive" mechanism. Specifically, the authors argue that the learned spectral coefficient should not only be inspected through quantities such as spectral entropy, effective rank, sparsity, perturbation sensitivity, and head similarity, but should also be used to guide any intervention on the attention operator itself.

The paper introduces different possible interventions, and provides several theoretical guarantees. Specifically, the authors provide a bound on the change in the pre-SoftMax scores after thresholding small entries of $\Sigma$ and relate this bound to changes in the attention distribution. In addition, they observe that a thresholded diagonal spectrum uses only its surviving coordinates and derive a bound involving $\|\Sigma\|_2$ for score sensitivity and provide the spectral similarity as a possible indication of attention head redundancy.

In the experiments, the authors study one of the considered interventions related to spectral energy-retention pruning. For each attention head, they retain the smallest set of directions whose squared coefficients contain at least a fraction $\rho$ of the total squared spectral magnitude. Regarding the other directions, they mask them first and then remove them from the Q, K projections and diagonal modulation, while the value and output remain uncompressed.

The authors do their experiments on FashionMNIST, CIFAR-10, CIFAR-100 and Food-101, using four-block SVDA-ViT models. When they set $\rho=0.9$, the method removes approximately $23\\%-49\\%$ of score directions, which leads to approximately $3.8\\%-7.6\\%$ fewer parameters and $4.1\\%-9.5\\%$ fewer estimates MACs. Finally, the authors compare against two more baselines, a random removal and the removal of the largest-$\Sigma$ directions baselines and include a small sensitivity study over $\rho$ on CIFAR-10.

---


## Strengths

**Motivation.** I like the high-level motivation of the paper, and I agree that an interpretability mechanism becomes considerably more useful when it is able to support a verifiable intervention rather than only producing descriptive statistics. The question of whether SVDA's learned coefficients can identify removable dimensions is natural and relevant.

**Simplicity.** The energy-retention rule is straightforward and easy to understand. In addition, it provides a deterministic, head-specific pruning rule without introducing an additional pruning model. I appreciate that the authors distinguish simply masking from actual structural removal and attempt to verify that the compressed model behaves similarly to the masked model.

**Matched Controls.** I find the largest-$\Sigma$ control useful because it tests whether the ordering of the learned coefficients contains at least some functional information. Removing the largest coefficients leads to considerably worse performance than removing the smallest coefficients which suggests that the learned spectrum is not just decorative. The authors also acknowledge that their method is approximately tied with random pruning on the compact datasets and that the estimated MAC reduction does not imply real wall-clock improvement.

That being said my main concern with the paper is that it presents a broad framework for "prescriptive attention", but empirically validates only a narrow and relatively straightforward magnitude-pruning procedure. In my opinion, the current evidence is sufficient to motivate further investigation, but it is not yet sufficient to support the broader framing of the paper.

---

## Weaknesses

**Theory and Experimental Connection.** From my understanding, the primary theoretical result is about the threshold pruning with an absolute threshold $\tau$, whereas the actual algorithm in practice uses a cumulative energy-retention rule parameterized by $\rho$. The authors do not provide a result that can translate $\rho$ into a perturbation bound and this missing connection is important because $\rho$ controls only the relative energy that is removed. The absolute perturbations still depends on the overall scale of $\Sigma$, and, e.g., one could derive $|\Sigma-\Sigma_{\rho}|2 \leq |\Sigma-\Sigma{\rho}|_F \leq \sqrt{1-\rho},|\Sigma|_F$, and then use this through the score and SoftMax bounds. Without such a result, the rule used in the main experiment is loosely related to the theory.

**Validation of the Framework.** The authors discuss effective-rank control, robustness-oriented regularization and head-redundancy analysis, but non of these interventions is evaluated. The authors acknowledge this limitation themselves, and I believe that these components currently read like proposed future work rather than contributions of the present submission.

**Spectral Energy.** The quantity $\sigma_r^2$ is called directional spectral energy and is treated as an importance score. However, the actual contribution of direction $r$ to a score is given by $\sigma_r q_{ir}k_{jr}$, and its downstream importance also depends on the data distribution, the attention normalization, the value pathway and the loss. A large $\sigma_r$ is not necessarily important when the corresponding query and key activations are consistently small and two direction with the same $\sigma_r$ may have very different functional effects. Thereupon, $\sigma^2$ is the squared magnitude of a learned diagonal modulation coefficient, but is not automatically equivalent to singular-value energy of the attention matrix or to loss-based importance. I think that the authors should either justify this terminology more carefully or empirically establish the relationship.


**Theoretical Results** Theorem 1 is valid and is able to provide a useful basic sanity check. However, I find Propositions 2, 3 and 5 not providing much insight beyond what follows from the definitions. Specifically, Proposition 2 is the standard range of entropy-based effective rank, Proposition 3 states that a diagonal operator with $C$ non-zero entries depends on at most $C$ coordinates and Proposition 5 states that two heads are identical when all of their parameters are identical. Perhaps, these observations made in these propositions should be compressed accordingly to ensure space for additional empirical results that would solidify the connections between the theory and practice. Finally, I do not believe the proof of Proposition 1 sufficient as the smoothness on a fininte-dimensional space does not by itself establish a global Lipschitzness on a non-compact domain. SoftMax does have a bounded Jacobian under the right norms, but the authors should state the norm pair, and provide an explicit constant and prove the result instead of introducing an arbitrary constant.


**Structural Realization.** I believe this point is a quite important technical issue. I understand that if the compressed model simply selects the retained coordinates from the already normalized $Q$ and $K$, retains the original $1/\sqrt{d_k}$ scaling, and removes coordinates whose masked coefficients are zero, then the masked and compressed matrices should be mathematically identical (up to floating-point error). However, it seems like the provided logit differences and prediction agreement do not verify this on Food-101. I think that this suggests that the compressed implementation somehow modifies the operator. If that is indeed the case, the compressed model is not an exact structural implementation of the masked operator and the theoretical result for the masked model does not directly follow. Thereupon, the exact compressed score equation needs to be states and the source of the non-zero discrepancy needs to be explained.


**Statistical Evidence.** The authors only present means over three seeds, but do not include standard deviations or (ideally) confidence intervals. The authors also claim that accuracy is preserved "within experimental noise", but this is not supported by the presented statistics. This is relevant because the differences between SVDA-guided and random prunin are close to zero on FashionMNIST, CIFAR-10 and CIFAR-100, something that leads to my next issue.


**Tied Performance with Random Control.** As shown in the results, at the conservative pruning levels that are considered by the authors, SVDA-guided pruning performs approximately the same as matched random pruning on three of the four datasets and is only 0.19 percentage points better on Food-101. On CIFAR-10, the advantage over random pruning is between 0.02 and 0.06 percentage points across the three values of $\rho$. In contrast, indeed, removing the largest coefficients is harmful, and this establishes that the largest coefficient should not be removed. However, this does establish that the proposed low-energy ordering provides a useful pruning criterion beyond the redundancy of these models. Finally, it is unclear to me whether each random result averages over multiple independently samples masks. A single random mask per trained model would be an unstable control. The paper should report the distribution over many matched random masks.

**Limited Settings.** I find the experimental settings of the experiments limited. FashionMNIST and CIFAR are more like sanity checks, and unfortunately are not too convincing for evaluating modern transformer compression. On the other hand, Food-101 provides larger inputs and more tokens, but the experiment still uses a four-block model trained from scratch, with an accuracy which, as acknowledged, is not competitive. The current experiments therefore establish that the procedure works on small SVDA models, but they do not show that meaningful redundancy emerges in pretrained or practically relevant transformers.

**Comparison with Related Pruning Methods.** In the related-work, the authors mention general head and token pruning, but the experiments do not compare against such structured pruning methods (e.g. see X-Pruner, UVC, NViT and MDP). I believe this is an important omission. The proposed method does not need to outperform every alternative, but it needs to explain more clearly what is different in practice and compare against at least a few matched alternatives.

**Additional Comments:**

- From my understanding the method does not fix $Q$, $K$, and $V$ and training only $\Sigma$. The method appears to train the full SVDA model and then prune it, but this should be stated explicitly.

- The use of the term "spectral energy" can be confusing. I think that the authors need to distinguish the squared diagonal modulation coefficients from the singular values of the data-dependent attention-score matrix. I suggest using "coefficient energy" or explicitly justifying the spectral terminology.

- The relationship between $\rho$ and the number of retained directions follows directly from the definition of the energy-retention rule. Thereupon, the CIFAR-10 sensitivity study primarily verifies the implementation. In my opinion, to demonstrate that $\rho$ is a meaningful parameter, the paper needs a broader accuracy-compression curve and comparisons with alternative ranking criteria.

- The prediction-agreement metric can be misleading, as two models can have identical accuracy while disagreeing on different examples. I recommend reporting the confusion between correct-to-incorrect and incorrect-to-correct changes.

- I think it is important to clarify how relative logit discrepancy is aggregated. It is not clear whether Equation (46) is computed per example and averaged, per batch, or over all concatenated test logits.

- Table 1 and the CIFAR-10 sensitivity table report slightly different values. It would be helpful to clarify whether these come from different experimental runs.

- Please use \eqref{} consistently when referring to equations. For example, the discussion surrounding Equations (22)–(27) currently uses plain equation numbers inconsistently.

**Audience:**

Yes

**Audience Explanation:**

The topic could be interesting for research on interpretability, pruning, or spectral parameterizations. Specifically, it is interesting to see whether a learned diagonal modulation can serve as a useful signal for removing query-key dimensions. There is also a meaningful difference between using these coefficients only as diagnostic and using them to build operators.

Even the current empirical finding could still be useful if it is stated more carefully. The learned coefficients seem to identify directions whose removal is especially damaging. At more conservative compression levels, however, removing the smallest-coefficient directions performs about the same as random pruning. Thereupon, it would be valuable to understand when the learned spectrum contains information beyond what one could expect from overparameterization.

I do not think that the topic is uninteresting, but, I believe that the current manuscript does not provide enough evidence to support its conclusions.

**Broader Impact Concerns:**

I do not identify a major ethical concern that would require a separate Broader Impact Statement. The work studies attention analysis and model compression on standard image-classification datasets.

**Claims And Evidence:**

No

**Claims Explanation:**

In my opinion, the current results are not strong enough to support the paper. First, the theoretical analysis is fairly elementary and does not directly correspond to the cumulative energy-retention criterion used in the experiments. Secondly, the relationship between the masked and the compressed operator is clear enough. I think this is important because the provided results suggest that the two are not mathematically equivalent. Third, at the tested budgets, the proposed method performs roughly on par with the random baseline. Finally, although the paper proposes several possible interventions, only one is evaluated empirically, and the experiments are limited to relatively small models and a narrow set of classification tasks.

Given the current state of the paper, mayhaps one claim that could be supported by the current results, is that across the four considered settings, removing low-score coordinates at conservative budgets preserves average accuracy and appears safer than removing the highest-score ones. However, even this conclusion would require a more complete analysis and a precise definition of the compressed operator.

**Requested Changes:**

### Changes Critical for Acceptance

**Precisely Define the Score-Compressed Operator**

The paper needs to provide the exact equation used for the score-compressed attention head. In particular, the authors should clarify:

* whether the query and key vectors are normalized before or after selecting the retained coordinates;

* whether normalization is recomputed in the reduced space;

* whether the denominator remains $\sqrt{d_k}$ or changes to $\sqrt{k_h}$;

* whether the compressed model is fine-tuned after it is constructed.

In addition, the authors should then either prove that the masked and compressed operators are equivalent or give a bound on the error introduced when converting one into the other. The reported logit differences and prediction disagreements suggest that the two implementations are not exactly equivalent, so these discrepancies should be explained rather than only reported.

**Connect the Theory to the Rule in Eq. (45)**

Since the theory is presented as a central part of the paper, it should directly analyze the rule that is used in the experiments. As mentioned in the weaknesses, a starting point would be to bound the removed Frobenius and spectral norms in terms of the discarded fraction of squared coefficient magnitude, and then propagate this bound to the attention scores and attention distributions. The scale of $|\Sigma|_F$ or $|\Sigma|_2$ also matters. Reporting these quantities across layers, datasets, and random seeds would help show whether the same value of $\rho$ has a comparable meaning across different models.

**Correct Proposition 1**

Proposition 1 should provide an explicit softmax Lipschitz bound under clearly stated input and output norms. The current justification, which relies on smoothness in finite dimensions, is not enough to establish a global constant. The revised statement should either give an explicit value of $C_{\mathrm{sm}}$ or remove the unspecified constant and cite or prove an appropriate softmax perturbation result.

**Report Statistically Meaningful Results**

The experiments should report the accuracies of the original, masked, and compressed models, rather than only relative changes and should also include:

* means and standard deviations or confidence intervals;

* the individual result from each seed;

* more than three seeds where the computational cost is reasonable.

The phrase "within experimental noise" should only be used after the magnitude of that noise has been quantified. If the goal is to claim that two methods are practically equivalent, an equivalence test with a chosen tolerance would be more convincing than simply observing a small difference between their mean accuracies.

**Strengthen the Random-Matched Control**

For each trained model and pruning budget, the authors should evaluate a sufficiently large number of independently sampled random masks. The resulting distribution should be reported, rather than relying on what appears to be a single random realization.

This would help to determine whether the proposed smallest-coefficient rule performs meaningfully better than random pruning, rather than merely falling somewhere inside the random distribution.

The largest-$\Sigma$ control is still useful, but it should be described as a destructive-ordering sanity check. It is not, by itself, a strong competitive pruning baseline.

**Compare With a Smaller Model Trained From Scratch**

A width-matched model with reduced query/key score dimensions from the beginning would be an important baseline. If such a model reaches the same or better accuracy, it becomes less clear why one should first train an SVDA model and then compress it using the learned spectrum. The authors should also consider a standard attention model with one learnable gate per dimension.

**Broaden the Experiments**

The current experiments appear too limited to support a general claim about Transformer compression or prescriptive attention. To support the broader framing, the paper should include at least one standard pretrained ViT backbone, such as DeiT or Swin, along with an ImageNet-scale evaluation. A non-vision experiment would also help support the more general Transformer claims.

**Validate the Interpretation of $\sigma_r^2$**

The paper should directly test whether the proposed "spectral energy" is actually related to functional importance. Useful analyses would include:

* the correlation between $\sigma_r^2$ and the loss or accuracy change caused by removing direction $r$;

* correlations with activation-weighted and gradient-based importance measures;

* layer-wise and head-wise pruning ratios;

* examples where coefficient magnitude succeeds or fails as an importance measure.

**Validate Other Proposed Interventions**

Effective-rank targeting, robustness regularization, and head-redundancy analysis are presented as contributions, but none of them is evaluated empirically. The authors should provide experiments that test these proposed uses. Otherwise, they should be moved to a shorter future-work discussion.

**Clarify the Training Procedure and Reproducibility Details**

It should be specified whether $Q$, $K$, $V$, the output projections, and $\Sigma$ are all trained jointly. My reading is that they are, rather than $Q$, $K$, and $V$ being fixed while only $\Sigma$ is learned, but this should be stated clearly.

The paper should also report:

* the optimizer and learning-rate schedule;

* the data augmentation procedure;

* weight decay;

* the initialization of $\Sigma$;

* any orthogonality or spectral regularization;

* the model-selection procedure;

* whether the test set was used in any way to choose $\rho$;

* the complete formula used to calculate MACs.

---

### Changes That Would Strengthen the Work

**Shorten and Refocus the Presentation**

The paper would benefit from reducing some of the repeated motivation and high-level claims. For example, the contributions appear to be stated twice in the introduction and could be combined into a single, more compact list. The terms "prescriptive" and "controlled intervention" could also be defined once and then used consistently. More broadly, Sections 1, 3, 3.8, 4, and 6 repeat several of the same ideas.

Propositions 3 and 5 could likely be replaced with shorter observations, while some of the more elementary derivations could be moved to the appendix. The resulting space would be better used for empirical comparisons, uncertainty estimates, and visualizations. In its current form, the amount of theoretical material makes the scope of the contribution appear broader than what is supported by the experiments.

The references and appendix should also be reorganized. At present, the References heading appears before all of the appendix tables have been presented, and Table 8 appears after the reference list begins. The authors should either place the complete reference list before the appendix or move all appendix material before the references.

Overall, a shorter and more carefully organized presentation would make the paper’s actual contribution easier to identify.